# Conformational tuning improves the stability of spirocyclic nitroxides with long paramagnetic relaxation times

Mateusz P. Sowiński [1], Sahil Gahlawat [1,2], Bjarte A. Lund [1,4], Anna-Luisa Warnke [1,4], Kathrin H. Hopmann [1], Janet E. Lovett [3] & Marius M. Haugland [1]✉

Nitroxides are widely used as probes and polarization transfer agents in spectroscopy and imaging. These applications require high stability towards reducing biological environments, as well as beneficial relaxation properties. While the latter is provided by spirocyclic groups on the nitroxide scaffold, such systems are not in themselves robust under reducing conditions. In this work, we introduce a strategy for stability enhancement through conformational tuning, where incorporating additional substituents on the nitroxide ring effects a shift towards highly stable *closed* spirocyclic conformations, as indicated by X-ray crystallography and density functional theory (DFT) calculations. *Closed* spirocyclohexyl nitroxides exhibit dramatically improved stability towards reduction by ascorbate, while maintaining long relaxation times in electron paramagnetic resonance (EPR) spectroscopy. These findings have important implications for the future design of new nitroxide-based spin labels and imaging agents.

---

[1] Department of Chemistry, UiT The Arctic University of Norway, 9037 Tromsø, Norway. [2] Hylleraas Center for Quantum Molecular Sciences, UiT The Arctic University of Norway, 9037 Tromsø, Norway. [3] SUPA, School of Physics and Astronomy and BSRC, University of St Andrews, North Haugh, St Andrews KY16 9SS, UK. [4] These authors contributed equally: Bjarte A. Lund, Anna-Luisa Warnke. ✉email: marius.m.haugland@uit.no

Nitroxides (also known as aminoxyl or nitroxyl radicals) are suited for innumerable applications across the chemical sciences[1,2]. These stable organic radicals are, for example, used as oxidants in synthetic organic chemistry[3], mediators for radical polymerization[4–7], redox-active components for materials science[8,9], and potential therapeutic agents[10]. Nitroxides have also found a unique place of significance in spectroscopy. Due to the localized radical distribution, small molecular size, and readily tunable properties through structural modification, nitroxides are highly suitable for use as observable probes in electron paramagnetic resonance (EPR) spectroscopy[11–13] and EPR imaging[14]. Moreover, nitroxides are used in nuclear magnetic resonance (NMR) spectroscopy as dynamic nuclear polarization (DNP) transfer agents[15,16], paramagnetic relaxation enhancement (PRE) probes[17], and as contrast agents in magnetic resonance imaging (MRI)[18,19].

Within the field of EPR spectroscopy, double electron-electron resonance spectroscopy (DEER), also known as pulsed electron double resonance (PELDOR), is a powerful tool for the investigation of biomolecular structure[20]. DEER experiments enable the measurement of nanometer-scale distances and angles between two spin centers (referred to as spin labels), which have been incorporated into biomolecular systems via site-directed spin labeling (SDSL)[13,21,22]. The resolution of DEER measurements is, among other things, influenced by the temperature-dependent relaxation parameter $T_m$, the phase memory time, which should be as high as possible[22].

For imaging and for modern spectroscopy applications, moreover, nitroxides should possess a sufficiently high aqueous solubility and stability towards the reducing environments typically found within cells[22]. In the presence of reducing agents such as ascorbate or glutathione, nitroxides are known to undergo reduction to diamagnetic hydroxylamines, which are not detectable by EPR[23]. Several structural factors influence nitroxide stability towards reduction (Fig. 1a). The size and saturation of the nitroxide ring has a significant impact: saturated five-membered pyrrolidinyl nitroxides are more stable than the unsaturated five-membered pyrrolinyl nitroxides, while six-membered piperidinyl nitroxides are more prone to reduction[22]. Electronic factors also influence the reduction resistance of nitroxides: electron-donating substituents enhance stability, while electron-withdrawing moieties decrease it[24]. Finally, the α-substituents shielding the paramagnetic center are a crucial factor. Replacing the most common methyl substituents with spirocyclohexyl groups results in a slight increase in the stability in reducing environments[24]. However, stability is significantly enhanced by more conformationally mobile ethyl substituents, which effectively occupy a larger spatial volume surrounding the spin center[25,26].

Beyond their impact on reduction stability, α-substituents also have a critical influence on $T_m$. The rotation of methyl groups is a significant contributor to the relaxation of nitroxides, especially at temperatures above 70 K[27]. Thus, the commonly used tetramethyl-substituted nitroxides require the use of cryogenic temperatures for DEER. This effect is also present in tetraethyl-substituted nitroxides, where a lower energy barrier for methyl rotation leads to increased relaxation rates at even lower temperatures[28]. In spirocyclic systems, however, the lack of methyl groups significantly enhances $T_m$, even at higher temperatures[29,30]. Thus, the design of nitroxides for in-cell applications is often a compromise between high reduction stability and long $T_m$.

The conformation of spirocyclohexyl nitroxides is an important consideration: it has been proposed that such scaffolds exist as an equilibrium between *open* and *closed* conformations, where the former is preferred in solution (Fig. 1b)[30,31]. *Closed* conformations are, however, expected to have higher stability in reducing environments due to more effective steric shielding of the nitroxide. Previous investigations have demonstrated that disubstituted five-membered pyrrolidinyl nitroxides (e.g., **1**, Fig. 1c) can adopt *semi-open* conformations with improved reduction stability[32]. Moreover, six-membered spirocyclohexyl nitroxides can be forced to adopt more stable *semi-open* or even *closed* conformations by the introduction of substituents on the cyclohexane rings (e.g., **2**)[31,33,34].

Here, we report that structural modification of the nitroxide rings itself can cause a conformational shift to provide nitroxides in fully *closed* conformations (Fig. 1d). The additional steric shielding obtained through this strategy results in a dramatic increase in the stability of these nitroxides towards reduction by ascorbate. Moreover, the beneficial $T_m$ of spirocyclic systems is largely preserved, resulting in a scaffold that finds a balance between improved stability and beneficial relaxation properties. These findings are of great significance for the future design of new nitroxide-based spin labels and imaging agents.

## Results and discussion

**Synthesis, X-ray crystallography, and conformational analysis.** Our initial strategy for the synthesis of substituted spirocyclohexyl piperidine nitroxides relied on classical enolate chemistry (Fig. 2a). Spirocyclohexyl piperidinone **3** was prepared from the commercially available 2,2,6,6-tetramethylpiperidinone according to a literature procedure[35]. Deprotonation of **3** with lithium diisopropylamide (LDA), followed by alkylation with methyl iodide, resulted in the mono-methylated product **4** in moderate yield. Subsequent oxidation with *m*-chloroperbenzoic acid (*m*-CPBA) provided nitroxide radical **5** in 84% yield. As radical **5** is crystalline, we investigated its three-dimensional structure by X-ray crystallography to reveal the conformational effect of a single substituent on the nitroxide ring. The solid-state structure (Fig. 2b) indicates that the methyl group adopts an axial orientation, with the nitroxide ring preferring a twist-boat conformation. Indeed, twist-boat conformations have also previously been observed in crystal structures of spirocyclic piperidinyl nitroxides[29,36]. The twist-boat conformation permits the spirocyclic scaffold to maintain an overall *open* arrangement, despite the presence of the additional methyl substituent.

We next aimed to force the spirocyclohexyl moieties into *closed* conformations by further increasing the steric pressure through additional α-functionalization of ketoamine **4**. However, this highly sterically hindered substrate proved unreactive towards further alkylation, either upon deprotonation with LDA and treatment with various alkylating agents (e.g., MeI, Me₂SO₄), or

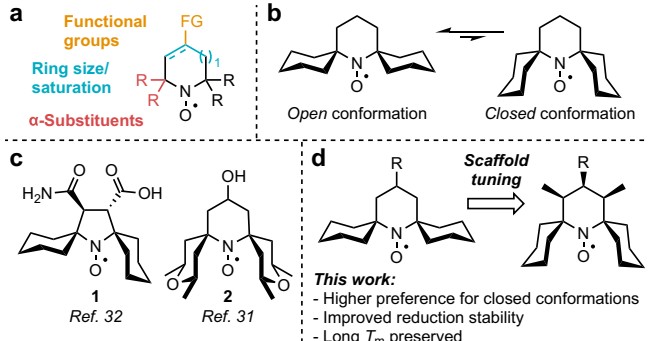

**Fig. 1 Context and motivation for this work. a** Structural factors affecting nitroxide stability. **b** Proposed equilibrium between *open* and *closed* conformations of spirocyclohexyl nitroxides. **c** Examples of conformational shifts in spirocyclic nitroxides[31,32]. **d** This work.

under Stork enamine alkylation conditions. Conversely, starting from ketoamine **3**, two methylene substituents were directly incorporated through condensation with formaldehyde (Fig. 2a)[37]. Although dienone **6** could be isolated, it polymerized in a matter of hours. Thus, it was immediately subjected to hydrogenation to afford dimethyl spirocyclohexyl ketoamine **7** in excellent yield as a mixture of diastereomers (*dr* 1:1). Subsequent oxidation with *m*-CPBA resulted in two diastereomers of dimethylated nitroxide **8**, which was recrystallized to enable structural investigation by X-ray crystallography. We obtained a solid-state structure for the *trans*-diastereomer of **8**, where the nitroxide ring adopts a chair conformation with one methyl substituent in an axial and the other in an equatorial orientation. The equatorial methyl group induces a conformational change of the proximal spirocyclohexyl moiety, resulting in a *semi-open* arrangement.

The two diastereomers of **8** were not readily separable by chromatography; moreover, we were concerned about possible diastereomeric interconversion via keto-enol tautomeric equilibria.

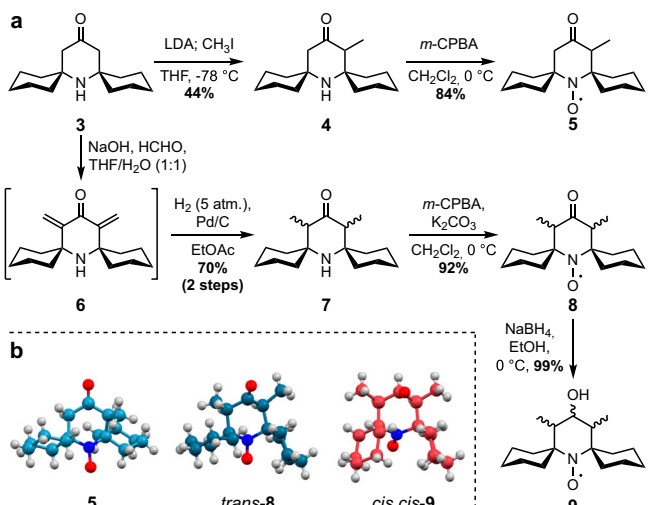

**Fig. 2 Synthetic route and solid-state structures of nitroxides. a** Synthesis strategies for obtaining substituted spirocyclohexyl nitroxides **5**, **8**, and **9**. **b** Crystal structures of solid-state nitroxides **5**, *trans*-**8**, and *cis,cis*-**9** obtained by X-ray crystallography.

Thus, we proceeded to reduce ketone **8** to alcohol **9**, which was obtained in excellent yield as a mixture of two diastereomers (*dr* 1:1), which were separable by column chromatography. The *cis,cis*-diastereomer of **9** was a crystalline solid, with the relative stereochemistry confirmed by X-ray crystallography (Fig. 2b). To our delight, *cis,cis*-**9** exhibits a *closed* conformation, presumably due to the combined steric pressure exerted by the two equatorial methyl substituents. The nitroxide ring adopted a chair conformation, placing the alcohol functionality in an axial position.

**Kinetic studies—stability toward reduction.** The stability of nitroxides towards reduction was measured by continuous-wave (CW) X-band EPR spectroscopy as a function of decay of the characteristic triplet signal in the presence of sodium ascorbate (1.4 equivalents). These conditions permitted the observation of meaningful decay rates at nitroxide concentrations high enough to be reliably measured on a benchtop EPR spectrometer. The time-dependent decay curves of the novel nitroxides **5**, **8**, and *cis,cis*-**9** are shown in Fig. 3a. To confirm our hypothesis, these substituted nitroxides were compared with the non-methylated spirocyclohexyl nitroxides **10** and **11**. The reduction kinetics of the well-established tetramethyl-substituted TEMPOL (**12**) and tetraethyl-substituted TEEPOL (**13**), were included as benchmarks for the piperidinyl nitroxides that are least and most resistant towards reduction, respectively[25]. Second-order rate constants *k* were derived by fitting decay data to a second-order kinetic rate law (see the SI for more details), and are included in Fig. 3c. Deviations from ideal second-order behavior were apparent, especially for rapidly reducing nitroxides, for which reliable rate constants could not be extracted. The non-ideal reduction kinetics are consistent with previous findings that the mechanism of nitroxide reduction by ascorbate is reversible and proceeds through a complex mechanism[23].

The presence of functional groups is known to influence the reduction rate of nitroxides[25]. In line with previous studies, the presence of an alcohol vs a ketone functionality has a negligible impact on the rate of reduction for the conditions and nitroxide scaffolds examined in this work. It is known that spirocyclohexyl-substituted piperidinyl nitroxides show slightly higher stability towards reduction than tetramethyl-substituted nitroxides, but not higher than tetraethyl-substituted radicals[26,38,39], a trend that is also clear from the decay curves in Fig. 3a. However, we were

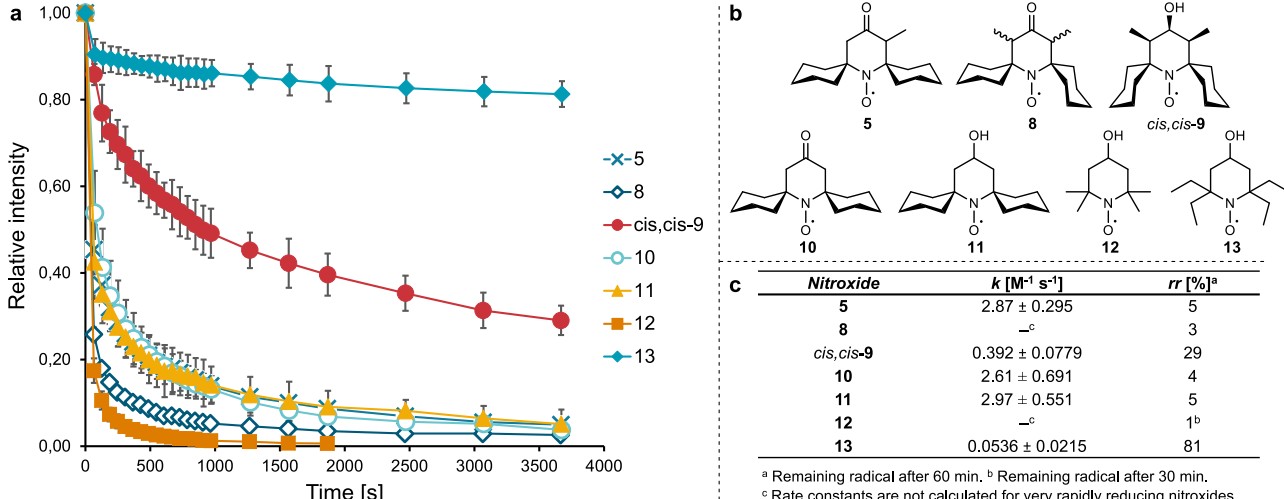

**Fig. 3 Kinetic decay studies of nitroxides. a** Second-order kinetic reduction curves of nitroxides (2 mM) with sodium ascorbate (1.4 eq.) in PBS buffer/DMSO (1:1) at 291 K. Data points are the average of three runs, with error bars representing 2 × standard deviation. **b** Structures of nitroxides for kinetic studies. **c** Second-order rate constants *k* [M⁻¹ s⁻¹] for 2 mM piperidinyl nitroxides, derived from decay curves by fitting to a second-order kinetic rate law. See the SI, Figs. S11, S12, and Table S1 for further details.

| Nitroxide | $k$ [M⁻¹ s⁻¹] | rr [%][a] |
|---|---|---|
| 5 | 2.87 ± 0.295 | 5 |
| 8 | —[c] | 3 |
| *cis,cis*-9 | 0.392 ± 0.0779 | 29 |
| 10 | 2.61 ± 0.691 | 4 |
| 11 | 2.97 ± 0.551 | 5 |
| 12 | —[c] | 1[b] |
| 13 | 0.0536 ± 0.0215 | 81 |

[a] Remaining radical after 60 min. [b] Remaining radical after 30 min.
[c] Rate constants are not calculated for very rapidly reducing nitroxides.

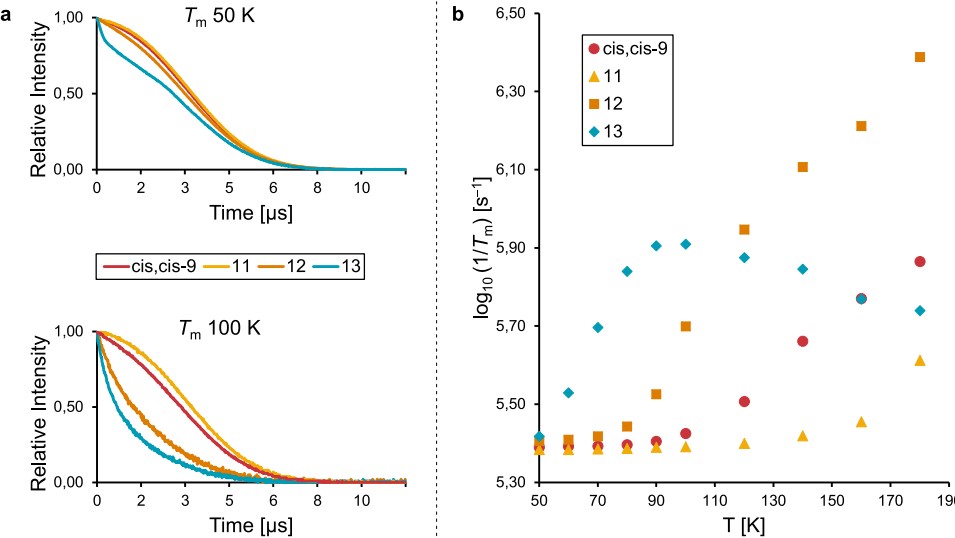

**Fig. 4 Relaxation time measurements. a** Spin echo decay curves for nitroxides *cis,cis*-**9**, **11**, **12,** and **13** at 50 and 100 K. **b** Temperature-dependent $T_m$ values for nitroxides *cis,cis*-**9**, **11**, **12**, and **13** measured by spin echo decay. The method for the extraction of $T_m$ values is described in the SI, Fig. S13.

pleased to observe a significant decrease of almost one order of magnitude in the rate of reduction of dimethylated spirocyclohexyl nitroxide *cis,cis*-**9** compared to non-methylated nitroxide **11**. This effect can be ascribed to the conformational effect on the spirocyclohexyl rings caused by the additional steric pressure from the two methyl substituents in **9**, causing the compound to prefer a *closed* conformation, similar to that observed in the solid-state X-ray crystal structure (Fig. 2b), that more effectively shields the radical from reducing agents. The reduction rate of mono-methylated ketone nitroxide **5**, however, does not differ significantly from that of the non-methylated version **10**. This is also in agreement with the *open* conformation adopted in the crystal structure of this compound (Fig. 2b). Surprisingly, the dimethylated ketone **8**, which was used as a mixture of diastereomers, is reduced more rapidly than all other spirocyclohexyl nitroxides. While the *trans*-diastereomer of **8** adopts a *semi-open* conformation in the solid state (Fig. 2b), the preferred conformation of the other diastereomer is unknown. Thus, at this point, the conformation of **8** cannot be related to its reactivity.

**Relaxation time measurements**. For potential applications as spin labels for DEER or polarization transfer agents in DNP NMR, nitroxides should also ideally exhibit sufficiently long relaxation times ($T_m$) to enable precise measurement of long distances, and efficient polarization transfer for signal enhancement, respectively[20,40]. It is well-known that the presence of methyl groups shortens $T_m$ due to methyl rotation, especially at temperatures above 70 K[27]. Although the requirement for using cryogenic temperatures that rely on liquid helium increases the operating costs of EPR spectrometers, contemporary instruments now minimize liquid helium consumption by the use of circulating liquid helium pumps[41]. Nevertheless, for reasons of cost and sustainability, it is desirable to preserve extended $T_m$ into temperatures that can be acquired using liquid nitrogen, especially for DNP NMR applications[42].

For $T_m$ measurements, nitroxides were dissolved in 1:1 PBS buffer/glycerol, in which concentrations of 50 µM were easily achieved. After flash freezing in liquid nitrogen, samples were cooled to 50 K and spin echo decay curves were recorded using pulsed Q-band EPR spectroscopy. The spin echo decays of nitroxide *cis,cis*-**9**, **11**, **12** and **13** at this temperature are shown in Fig. 4a. Dimethylated spirocyclohexyl nitroxide *cis,cis*-**9** has a

similar echo decay to the non-methylated spirocyclohexyl nitroxide **11**, which exhibits the longest $T_m$. Tetramethyl nitroxide **12** has a slightly shorter $T_m$ even at 50 K, but tetraethyl nitroxide **13** shows significantly more rapid relaxation. This has previously been ascribed to an increased local proton concentration in the tetraethyl substituents[43]. However, we note that spirocyclohexyl nitroxides also have an increased local proton concentration, and that the line shape of the echo decay of **13** suggests there are multiple mechanisms of spin echo dephasing in operation for the tetraethyl system at 50 K.

We next investigated $T_m$ across a range of temperatures. A plot of $T_m$ as a function of temperature (Fig. 4b) clearly shows how methyl rotation significantly increases relaxation in tetramethyl nitroxide **12** (from 80 K) and tetraethyl nitroxide **13** (from 60 K). The earlier onset of methyl rotation in tetraethyl nitroxides is explained by a lower energy barrier of methyl rotation in these systems[44]. It is not clear, however, why the $T_m$ apparently increases again, and becomes longer than the tetramethyl nitroxide **12** as the temperature rises above *ca.* 120 K. This unexpected behavior is in line with observations in our previous work, and suggests further complications connected with the use of tetraethyl nitroxides[45]. The origins of this effect will be further investigated in the future. Conversely, spirocyclohexyl nitroxides *cis,cis*-**9** and **11** exhibit low $T_m$ dephasing rates even at higher temperatures, with dimethylated spirocyclohexyl nitroxide *cis,cis*-**9** having relaxation rates similar to the non-methylated analog **11** up to *ca.* 100 K. Additional dephasing beyond this temperature is presumably driven by rotation of the methyl substituents on the nitroxide ring. The spin echo decay curves at 100 K (Fig. 4a) illustrate how distinct behavior of different methyl environments have a critical impact on $T_m$ at this temperature, which can be easily achieved using liquid nitrogen cooling. The diversity of the impact of methyl substituents on $T_m$ between *cis,cis*-**9** and **12** may be a function of the distance between the paramagnetic center and the methyl groups, or different rates of methyl rotation[44]. The alignment of the group with the anisotropic spin tensor of the electron may be an additional contributing factor.

**DFT calculations**. To assess the impact of the additional nitroxide ring substitution across the conformational equilibrium, a computational analysis using density functional theory (DFT, PBE0-D3BJ/def2-TZVPP[IEFPCM])[46–56] was performed. The robustness of computed results was checked by applying B3LYP-D3 for

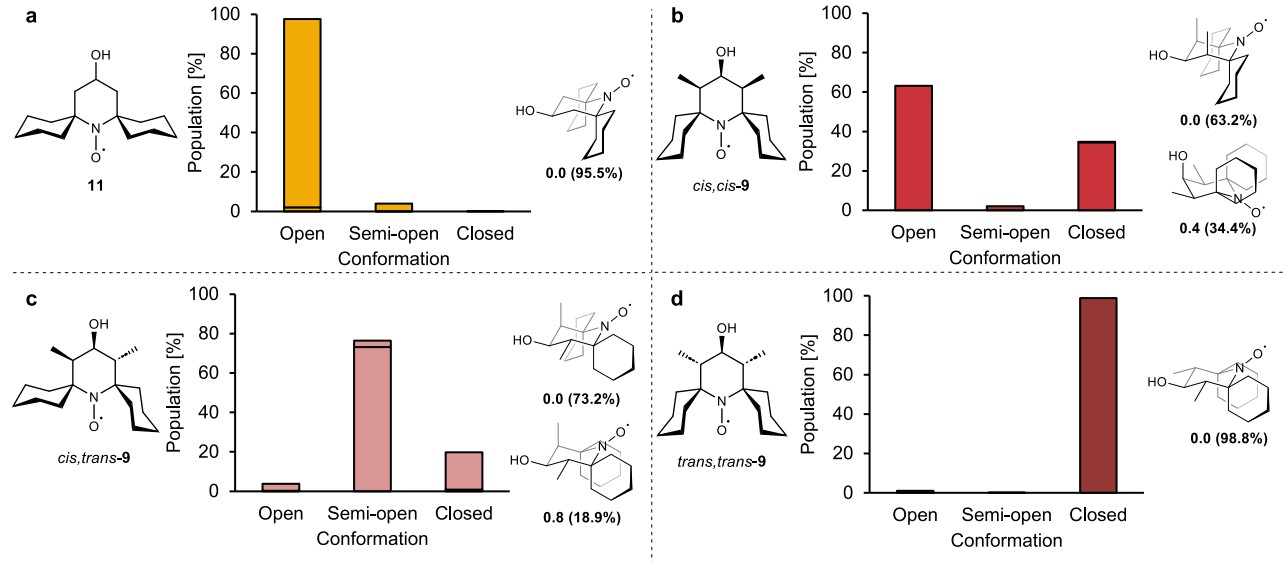

**Fig. 5 Computational investigation of the conformational distribution of alcohol nitroxides.** Distribution of calculated conformers in *open, semi-open*, and *closed* conformations for nitroxides **11** (**a**), *cis,cis-***9** (**b**), *cis,trans-***9** (**c**), and *trans,trans-***9** (**d**). The most highly populated conformers are shown for each nitroxide, along with the calculated Gibbs free energies (in kcal/mol, relative to the most stable conformation) and Boltzmann distribution (parentheses, in percentage) (298 K, PBE0-D3BJ/def2-TZVPP[IEFPCM, water])[46-56]. See the SI, Fig. S15, for all calculated conformations and energies.

comparison (SI, Fig. S14). First, Gibbs free energies and Boltzmann distributions were computed for *open, semi-open*, and *closed* conformations of alcohol nitroxides **11** and **9** (Fig. 5 and SI, Fig. S15). Conformers were identified where both the central nitroxide-containing ring and spirocyclohexyl substituents adopt chair conformations. Interconversion between these conformers corresponds to a ring-flip of either the nitroxide-containing heterocycle or the spirocyclohexyl rings, via intermediate twist-boat and chair structures. We expect the identified conformations to be freely interconverting in solution at room temperature; as the nitroxide functionality is approximately planar, nitroxides **11** and **9** are structurally comparable to highly sterically crowded cyclohexanones, which have an estimated energy barrier for conformational interconversion of 16.7 kcal/mol[57]. For the non-methylated nitroxide **11** (Fig. 5a), the *open* conformation with an equatorial hydroxyl group is favored over the corresponding *closed* conformation by 2.2 kcal/mol, in line with the expectation that this compound favors an *open* conformation. This contrasts with the dimethylated analog **9**, which can exist as the diastereomers *cis,cis-***9**, *trans,trans-***9**, and *cis,trans-***9**. For *cis,cis-***9** there are two low-energy conformations: a *closed* structure with equatorial methyl groups (as also observed in the solid-state crystal structure, Fig. 2b), and an *open* conformation with both methyl groups in axial positions (Fig. 5b). These structures are the predominantly populated conformations in solution, in an approximate ratio of 1:2. Notably, the two most stable conformers have very similar energies, resulting in a change in their order of relative stability when using B3LYP-D3 instead of PBE0-D3BJ (see SI, Fig. S14). Thus, the calculations support our hypothesis that methyl substituents at the 3- and 5-positions of the nitroxide ring exert a steric pressure that can increase the preference for the *closed* conformation.

The calculations also show that the *relative* stereochemical configuration of the ring substituents is important. The diastereomer *cis,trans-***9** (Fig. 5c) mostly populates *semi-open* and *closed* conformations, with the former, lowest-energy structure demonstrating that an equatorial methyl group is more effective at forcing the spirocyclohexyl ring to adopt a *closed* position than an axial group, for nitroxide rings in a chair conformation. In line with this, the last possible diastereomer, *trans,trans-***9**, displays a strong preference for the *closed* over the

*open* conformation, by 5.0 kcal/mol, if all nitroxide ring substituents are equatorial and 2.7 kcal/mol if substituents are axial (Fig. 5d and SI, Fig. S15). The increased preference for *closed* conformations of these other diastereomers of **9** suggests that they may be even more stable towards reduction than *cis,cis-***9**.

The calculated conformational energies and Boltzmann distributions of ketone nitroxides **10**, **5**, and **8** present a more complicated picture (Fig. 6 and SI, Fig. S16). Due to reduced torsional strain, six-membered rings with two sp²-hybridized centers in a 1,4-relationship can readily access low-energy twist-boat conformations[58]. Reduced strain in the transition states reported for conformational interconversion further indicates that such systems will rapidly interconvert at room temperature[59]. Indeed, the non-methylated ketone nitroxide **10** preferentially adopts an *open* twist-boat conformation, which is favored over a chair conformation by 1.7 kcal/mol (Fig. 6a and SI, Fig. S16). In mono-methylated nitroxide **5** (Fig. 6b), a small energetic preference is found for an *open* twist-boat conformation that places the methyl substituent in an axial orientation, as observed in the solid-state crystal structure of this compound (Fig. 2b). Nitroxide **5**, however, also populates several other low-energy *open* and *semi-open* structures, but the steric pressure exerted by a single methyl substituent is apparently insufficient to induce any significant population of *closed* conformations. The *trans*-diastereomer of dimethylated nitroxide **8** prefers a *semi-open* chair conformation, as observed in the X-ray structure (Fig. 2b), but can also access two *closed* structures only 0.5 kcal/mol higher in energy (Fig. 6c). For the *cis-***8** diastereomer, a preference is found for a *semi-open* twist-boat conformation, but this compound also significantly populates a *closed* chair conformation that is only 0.2 kcal/mol higher in energy (Fig. 6d). Overall, the calculated conformational energies for nitroxides **5**, **8**, and **10** indicate that the introduction of methyl substituents does increase the occupancy of *semi-open* and *closed* conformations also for ketone spirocyclohexyl nitroxides, although this does not translate into increased stability towards reduction (cf. Fig. 3a). A possible explanation may be found in a higher thermodynamic electron affinity of **5** and **8** (*vide infra*).

To assess any thermodynamic contribution to reduction stability, we next calculated nitroxide electron affinities (EAs).

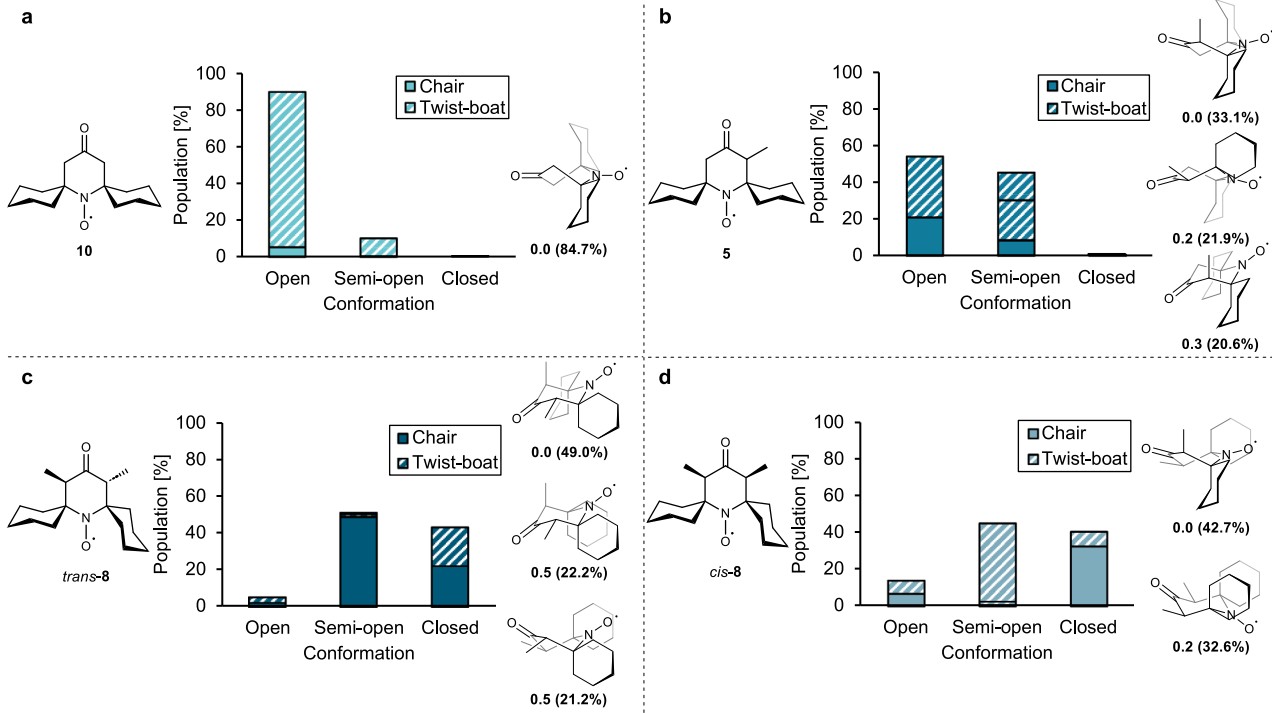

**Fig. 6 Computational investigation of the conformational distribution of ketone nitroxides.** Distribution of calculated conformers in open, semi-open, and closed conformations for nitroxides **10** (**a**), **5** (**b**), *trans*-**8** (**c**), and *cis*-**8** (**d**). The most highly populated conformers are shown for each nitroxide, along with the calculated Gibbs free energies (in kcal/mol, relative to the most stable conformation) and Boltzmann distribution (parentheses, in percentage) (298 K, PBE0-D3BJ/def2-TZVPP[IEFPCM, water])[46–56]. See the SI, Fig. S16, for all calculated conformations and energies.

Energies were calculated for optimized geometries of all possible conformations of the hydroxylamine anions formed by single-electron reduction of nitroxide radicals (SI, Fig. S17), and the adiabatic EA was taken as the energy difference between the most stable conformation of the nitroxide (Figs. 5, 6) and the most stable conformation of the hydroxylamine anion (SI, Fig. S17) for each compound. The EAs (Fig. 7) are generally higher for ketone nitroxides compared to the alcohols. This may be due to an electronic contribution from the electron-withdrawing carbonyl groups. For the alcohol nitroxides, dimethylated spirocyclohexyl nitroxide *cis,cis*-**9**, and the non-methylated analog **11** have identical EAs, which supports our assumption that the stabilization achieved by the introduction of methyl groups is a kinetic effect. By comparison, tetramethyl-substituted nitroxide **12** has a slightly higher apparent EA, while the EA of tetraethyl analog **13** is considerably lower. We ascribe this trend to the electronic repulsion between the hydroxylamine anion and the α-substituents, which is reduced in **12** and increased in **13**, and constitutes a thermodynamic factor that disfavors the reduction of nitroxides. The ketone nitroxide *cis*-**8**, however, has a significantly higher EA than all other nitroxides, which counteracts the presumed stability in more highly populated *semi-open* and *closed* conformations. This can explain the experimental finding (Fig. 3a) that nitroxide **8** (as a mixture of both diastereomers) is reduced more rapidly than the other spirocyclohexyl nitroxides.

## Conclusion

In this work, we have successfully developed a new and reliable synthetic route to obtain a series of novel 3,5-substituted spirocyclohexyl piperidinyl nitroxides. Investigation of the conformations of these compounds by X-ray crystallography and DFT calculations confirmed our hypothesis that the introduction of additional substituents on the nitroxide ring can induce spirocyclohexyl nitroxides to favor *closed* conformations. Our results indicate that only methyl substituents in an equatorial position on chair conformations of the central nitroxide ring can maximize the steric pressure to fully enforce a *closed* conformation of the nearby spirocyclohexyl groups. As indicated by kinetic decay studies with CW-EPR, this conformational biasing significantly increases the stability of spirocyclohexyl nitroxides towards reduction by ascorbate. Moreover, the beneficial extended phase memory times ($T_m$) exhibited by spirocyclohexyl nitroxides in pulsed EPR are generally preserved despite the introduction of methyl substituents on the nitroxide ring, thereby striking an excellent compromise between reduction stability and extended $T_m$ in these scaffolds. These balanced properties are of great potential utility in nitroxides intended for applications such as biocompatible spin labels, imaging probes, or DNP NMR polarization transfer agents.

## Methods

General methods and materials are described in the Supplementary Information (Page 2, SI).

**Chemical synthesis**. Detailed synthetic procedures and characterization data for all nitroxides and intermediates are described in the Supplementary Information.

**X-ray crystallography**. Crystals of compounds **5**, **8**, and *cis,cis*-**9** were grown from diethyl ether by the slow evaporation method. Crystals were glued to glass fibers and mounted on the goniometer. Data were collected using a D8 Venture system with a Cu-anode d ($\lambda = 1.54178$ Å). See the SI, Figs. S1–S3 for the ellipsoids.

**Electron paramagnetic resonance (EPR) spectroscopy**. Room temperature CW-EPR measurements were acquired using an X-band Affirmo microESR Benchtop EPR Spectrometer (now marketed as the Bruker microESR). See the SI, Figures S4-S10 for X-band CW-EPR spectra of the nitroxides. Bruker Elexsys E580 with high powered (150 W) Q-band (34 GHz) and an ER 5106QT-2w cylindrical resonator was used for relaxation parameter investigations. Detailed parameters and procedures are described in the Supplementary Information.

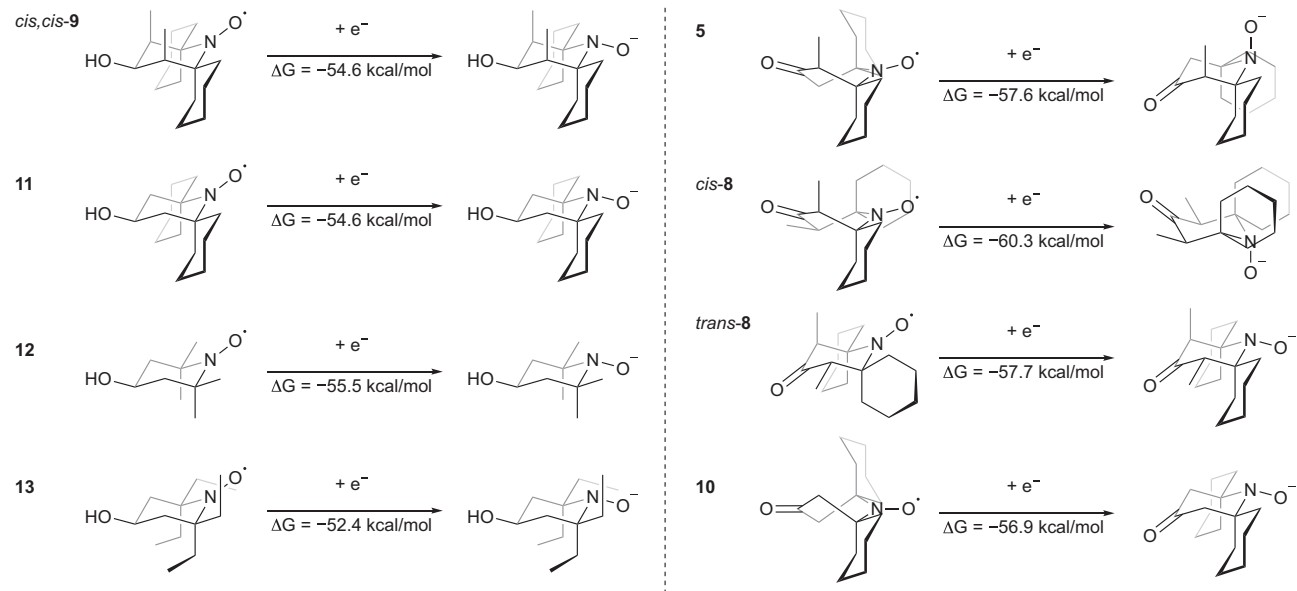

**Fig. 7 Computational investigation of nitroxide electron affinities.** Calculated adiabatic electron affinities (EAs) for nitroxides **5**, **8**, *cis,cis*-**9**, and **10**–**13** (298 K, PBE0-D3BJ/def2-TZVPP[IEFPCM, water])[46–56]. EAs were taken as the energy difference between the most stable nitroxide conformer and the most stable hydroxylamine anion conformer for each compound.

**Kinetic studies**. Sodium ascorbate solution in PBS buffer/DMSO (1:1 v/v) was used as a reduction agent. Ascorbate solution was added to the nitroxide solution and the peak height of the low-field line of the nitroxide triplet CW-EPR spectrum was measured as a function of time. Each kinetic run was repeated three times on the same day using the same ascorbate solution. Detailed procedures and data fitting are described in the Supplementary Information.

**DFT calculations**. All calculations were performed on complete molecular systems without any truncations using Gaussian16 (Revision B.01)[56]. The systems were fully relaxed, and no symmetry constraints were imposed. For the geometry optimizations, the DFT hybrid functional PBE0[49,50], which contains 25 percent Hartree–Fock exchange, was used along with the D3BJ[55] dispersion correction (additional calculations with B3LYP[47]-D3 provide similar trends, see Figure S14). The basis set used for geometry optimizations and frequency calculations was def2-TZVPP[46,48,52]. Solvation effects were included using the polarizable continuum model (IEFPCM)[54] with the parameters of water ($\varepsilon = 80$). Reported Gibbs free energies include thermal corrections computed at 298 K. See the SI, Fig. S18, for the optimized geometries. Hartree energies of all calculated conformers are shown in Table S2.

## Data availability

Copies of the NMR spectra are included in Supplementary Data 1 (Figs. S19–S46). The X-ray crystallographic coordinates for structures reported in this study have been deposited at the Cambridge Crystallographic Data Centre (CCDC), under deposition numbers 2214625, 2214626, and 2214627. These data can be obtained free of charge from The Cambridge Crystallographic Data Centre via www.ccdc.cam.ac.uk/data_request/cif. Moreover, CIF files and SCXRD reports for nitroxides 5, *trans*-8, and *cis,cis*-9 are included in Supplementary Data 2–7. Coordinates of all optimized geometries and their energies are given in Supplementary Data 8 (XYZ file). Additional raw data is available through the *DataverseNO* repository at https://doi.org/10.18710/UQMMZE[60].

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

## Acknowledgements

The authors thank Dr. D. G. Mazhukin for helpful suggestions for the preprint version of this work. M.M.H. and A.-L.W. thank the Tromsø Research Foundation and UiT Centre for New Antibacterial Strategies (CANS) for a start-up grant (TFS project ID: 18_CANS). K.H.H. and S.G. thank the Research Council of Norway (No. 300769) and Sigma2 (No. nn9330k and nn4654k), and the European Union's Horizon 2020 research and innovation program under the Marie Skłodowska-Curie grant agreement No 859910. J.E.L. thanks Drs Hassane El Mkami and Robert I. Hunter for technical assistance, the BBSRC (BB/T017740/1) and the Wellcome Trust (099149/Z/12/Z) for the Q-band EPR spectrometer, the Royal Society for a University Research Fellowship and Research Grant RG120645 for the benchtop spectrometer. B.A.L. thanks The Research Council of Norway for the Centre of Excellence and project grants (Grant Nos. 262695 and 274858).

## Author contributions

In descending order of degree of contribution. Conceptualization: M.M.H.; funding acquisition: M.M.H., J.E.L., and K.H.H.; methodology: M.M.H., J.E.L., K.H.H., S.G., and M.P.S.; investigation: M.P.S., S.G., A.-L.W., B.A.L., and J.E.L.; formal analysis: M.P.S., S.G., B.A.L., A.-L.W., M.M.H., J.E.L., K.H.H.; software: S.G., K.H.H., and J.E.L.; validation: M.P.S., S.G., B.A.L., M.M.H., K.H.H., and J.E.L.; data curation: M.P.S., S.G., and B.A.L.; visualization: M.P.S., M.M.H., and S.G.; writing—original draft: M.P.S., M.M.H., and S.G.; writing—review & editing: K.H.H., A.-L.W., J.E.L., and B.A.L.; project administration: M.M.H.; resources: M.M.H., J.E.L., and K.H.H.; supervision: M.M.H., K.H.H., and J.E.L.

## Funding

## Competing interests

The authors declare no competing interests.
