## [Peer Review File · Communications Chemistry]

Reviewers' comments:

Reviewer #1 (Remarks to the Author):

In their submission to Communications Chemistry entitled "Conformational tuning improves the stability of spirocyclic nitroxides with long paramagnetic relaxation times", Haugland and coworkers report on the design and use of 3,5-substituted spirocyclic nitroxides as probes with improved stability towards reduction by ascorbate. The introduction of methyl substituents in an equatorial position on chair conformations on the nitroxide ring maximise the steric bulk and favours closed conformations on the nearby spirocyclohexyl groups. The authors performed a detailed investigation of the conformations of these novel compounds by means of X-ray crystallography combined with DFT calculations to support the hypothesis. These compounds showed long relaxation times in EPR spectroscopy. The manuscript is well written and the results should be of interest to the scientific community and the broad readership of Communications Chemistry. The manuscript may be publishable, but it should be reviewed after major revisions.

- (1) A figure showing the optimized structures from the DFT calculations would be of help and would graphocally show the closed conformation of the nearby spirocyclohexyl groups. This can be included in Figure 4 and substitute the structures on the right of the graphics, as they are of poor quality.
- (2) Regarding the DFT calculations, a table with computed energies of the optimized structures (in hartree) should be included in the Supporting Information. Furthermore, imaginary frequencies for the transition states regarding the conformational interconversion should also be reported in the Supporting Information.
- (3) The quality of Figure 6 is poor and needs to be improved.

Reviewer #2 (Remarks to the Author):

Manuscript COMMSCHEM-23-0129 by Haugland and coworkers titled, "Conformational tuning improves the stability of spirocyclic nitroxides with long paramagnetic relaxation times", describes the synthesis of pyrimidine spirocyclic nitroxides that contain methyl groups on the nitroxide ring. The goal was to improve the stability of nitroxides towards reduction while keeping favourable electron relaxation rates that are important for biophysical applications. The authors investigated the effect of the methyl substitutions on rates of nitroxide reduction and electronic relaxation rates. Crystal structures of select nitroxides and DFT calculations were collected to provide a correlation between methyl substitution and the conformational equilibria. The authors showed that one derivative, the cis-cis 9, showed improved stability towards reduction, while keeping favourable rates of relaxation at the lower temperatures. Although it was disappointing to see that incorporation of the methyl groups into the cis-cis 9 increased the rate of relaxation relative to the unsubstituted spirocycle 11, it was still substantially lower than for the tetraethyl derivative 13 at low to intermediate temperatures. The work described in the manuscript is systematically and carefully carried out. Moreover, the paper is very well written. Given the current interest in free radicals that are resistant towards reduction for in-cell biophysical experiments, I recommend publication in Communications Chemistry after addressing the following minor points:

- Define the meaning of EPR and DFT in abstract.
- Line 9 in second para of second page: "The resolution of DEER measurements is, among other things, influenced...".
- Line 11 in fourth para on page 2: "...of methyl groups significantly enhances...".

- In several places in the text, an A, B, C or D is missing when referring to a scheme or a figure in the text (for example Scheme 1A).
- Fourth para on page 3, define CW.
- Figure 4/5 in the figure captions for Figures 4 and 5 are in incorrect order (Figure 5 comes before Figure 4).
- The minus signs by the deltaG in Figure 6 can hardly be seen. Make black.
- The references need editing in a number of places. Some titles have major words in caps, not others. Some journals are not abbreviated. Names of books missing. Names of journal articles missing.
- Subscripts and superscripts in legends of NMR spectra in the SI are not proper.

Point-by-point response to referees' comments

Reviewer #1:

In their submission to Communications Chemistry entitled "Conformational tuning improves the stability of spirocyclic nitroxides with long paramagnetic relaxation times", Haugland and coworkers report on the design and use of 3,5-substituted spirocyclic nitroxides as probes with improved stability towards reduction by ascorbate. The introduction of methyl substituents in an equatorial position on chair conformations on the nitroxide ring maximise the steric bulk and favours closed conformations on the nearby spirocyclohexyl groups. The authors performed a detailed investigation of the conformations of these novel compounds by means of X-ray crystallography combined with DFT calculations to support the hypothesis. These compounds showed long relaxation times in EPR spectroscopy. The manuscript is well written and the results should be of interest to the scientific community and the broad readership of Communications Chemistry. The manuscript may be publishable, but it should be reviewed after major revisions.

We thank the referee for their supportive comments and helpful suggestions about our manuscript, and are pleased to provide a detailed response to the particular points raised below.

(1) A figure showing the optimized structures from the DFT calculations would be of help and would graphically show the closed conformation of the nearby spirocyclohexyl groups. This can be included in Figure 4 and substitute the structures on the right of the graphics, as they are of poor quality.

We apologize for the poor resolution of Figure 4 (now Figure 5). We have provided a higher resolution version of all figures. We have also added figures of the DFT optimized geometries in the SI (Figure S18), corresponding to the structures in Figure 5. As part of the Supplementary Data, we have submitted an XYZ file that can be opened in the free software Mercury; the XYZ file displays the geometries of all structures easily to the reader. As we believe the simplified representations of the nitroxide conformers give a greater emphasis to the most important aspects of the conformational analysis, we have kept these representations in Figure 5 in the main manuscript.

(2) Regarding the DFT calculations, a table with computed energies of the optimized structures (in hartree) should be included in the Supporting Information. Furthermore, imaginary frequencies for the transition states regarding the conformational interconversion should also be reported in the Supporting Information.

We thank the referee for this suggestion. We have added all Hartree energies in the Supplementary Information (Table S2), as well as in the Supplementary Data file that contains the XYZ coordinates. Our analysis was based on thermodynamics, such that transition states for interconversion were not calculated, as this is beyond the scope for this paper. Thus there are no imaginary frequencies to be included.

(3) The quality of Figure 6 is poor and needs to be improved.

We apologize if Figure 6 (now Figure 7) did not have sufficient resolution in the previous version of the manuscript. We have provided a new Figure 7 with higher resolution, including as a separate image file.

Reviewer #2:

Manuscript COMMSCHEM-23-0129 by Haugland and coworkers titled, "Conformational tuning improves the stability of spirocyclic nitroxides with long paramagnetic relaxation times", describes the synthesis of pyrimidine spirocyclic nitroxides that contain methyl groups on the nitroxide ring. The goal was to improve the stability of nitroxides towards reduction while keeping favourable electron relaxation rates that are important for biophysical applications. The authors investigated the effect of the methyl substitutions on rates of nitroxide reduction and electronic relaxation rates. Crystal structures of select nitroxides and DFT calculations were collected to provide a correlation between methyl substitution and the conformational equilibria. The authors showed that one derivative, the cis-cis 9, showed improved stability towards reduction, while keeping favourable rates of relaxation at the lower temperatures. Although it was disappointing to see that incorporation of the methyl groups into the cis-cis 9 increased the rate of relaxation relative to the unsubstituted spirocycle 11, it was still substantially lower than for the tetraethyl derivative 13 at low to intermediate temperatures.

The work described in the manuscript is systematically and carefully carried out. Moreover, the paper is very well written. Given the current interest in free radicals that are resistant towards reduction for in-cell biophysical experiments, I recommend publication in Communications Chemistry after addressing the following minor points:

We thank the referee for their support and useful recommendations for our manuscript. Please see below for a detailed response.

- Define the meaning of EPR and DFT in abstract.

We have included a full definition of these abbreviations in the abstract.

- Line 9 in second para of second page: "The resolution of DEER measurements is, among other things, influenced...".

We have inserted the additional commas as requested.

- Line 11 in fourth para on page 2: "...of methyl groups significantly enhances...".

We have corrected this grammatical error as requested.

- In several places in the text, an A, B, C or D is missing when referring to a scheme or a figure in the text (for example Scheme 1A).

We thank the referee for noticing this omission. We have included specific references to figure panels throughout the manuscript.

- Fourth para on page 3, define CW.

We have included a definition of this abbreviation as requested.

- Figure 4/5 in the figure captions for Figures 4 and 5 are in incorrect order (Figure 5 comes before Figure 4).

We thank the referee for noticing this, and have renumbered the figures accordingly. Please note that due to style and formatting requirements, the figure numbers have now changed, and these figures are now numbered 5 and 6, respectively.

- The minus signs by the ΔG in Figure 6 can hardly be seen. Make black.

We apologize for the low resolution of the figures in the previous version of the manuscript. We have provided a new figure (now Figure 7) with higher resolution, in the manuscript and as a separate image file.

- The references need editing in a number of places. Some titles have major words in caps, not others. Some journals are not abbreviated. Names of books missing. Names of journal articles missing.

We thank the referee for pointing out the errors in the formatting of references. We have corrected the references where required.

- Subscripts and superscripts in legends of NMR spectra in the SI are not proper.

We thank the reviewer for finding this formatting error. We have corrected the formatting in legends for NMR spectra, as well as a couple of instances in the synthesis and characterization part of the Supplementary Information. Please note that the NMR spectra are now included as Supplementary Data file no. 1, according to the journal requirements.